# An InP-based vortex beam emitter with monolithically integrated laser

Juan Zhang[1], Changzheng Sun[1], Bing Xiong[1], Jian Wang[1], Zhibiao Hao[1], Lai Wang[1], Yanjun Han[1], Hongtao Li[1], Yi Luo[1], Yi Xiao[2], Chuanqing Yu[2], Takuo Tanemura[2], Yoshiaki Nakano[2], Shimao Li[3], Xinlun Cai[3] & Siyuan Yu[3]

Semiconductor devices capable of generating a vortex beam with a specific orbital angular momentum (OAM) order are highly attractive for applications ranging from nanoparticle manipulation, imaging and microscopy to fiber and quantum communications. In this work, an electrically pumped integrated OAM emitter operating at telecom wavelengths is fabricated by monolithically integrating an optical vortex emitter with a distributed feedback laser on the same InGaAsP/InP epitaxial wafer. A single-step dry-etching process is adopted to complete the OAM emitter, equipped with specially designed top gratings. The vortex beam emitted by the integrated device is captured and its OAM mode purity characterized. The integrated OAM emitter eliminates the external laser required by silicon- or silicon-on-insulator-based OAM emitters, thus demonstrating great potential for applications in communication systems and the quantum domain.

[1] Beijing National Research Centre for Information Science and Technology (BNRist), Department of Electronic Engineering, Tsinghua University, Beijing 100084, China. [2] Integrated Photonics Laboratory, Department of Electrical Engineering and Information Systems, Graduate School of Engineering, the University of Tokyo, Tokyo 113-8656, Japan. [3] State Key Laboratory of Optoelectronic Materials and Technologies, School of Electronics and Information Technology, Sun Yat-sen University, Guangzhou 510275, China. Theses authors contributed equally: Juan Zhang, Changzheng Sun.  Correspondence and requests for materials should be addressed to Y.L. (email: luoy@tsinghua.edu.cn) or to Y.N. (email: nakano@hotaka.t.u-tokyo.ac.jp) or to X.C. (email: caixlun5@mail.sysu.edu.cn)

O ptical vortex beams with a helical wave front are known to carry orbital angular momentum (OAM)[1]. The phase of an OAM-carrying beam varies with the azimuthal angle $\phi$, i.e., $E \sim \exp(jl\phi)$, where $l$ is the topological charge of the OAM state[1,2]. As a special form of space-division multiplexing (SDM), OAM offers an extra degree of freedom for information encoding, in addition to the already extensively exploited wavelength, amplitude, phase, and polarization. It holds the potential for enhancing the capacity and spectral efficiency of optical communication systems, as there are theoretically an unbounded number of orthogonal states[2,3]. In addition to the impressive demonstrations of their capacity in free-space and fiber communications[4–6], beams carrying OAM have also attracted substantial attention for myriad applications[7,8], including optical manipulation[9–11], imaging and microscopy[12,13], and quantum information[14,15].

Compared with the bulky optical components typically used for OAM beam generation, such as spiral phase plates[16], cylindrical lens-based mode converters[17], or forked holograms implemented with spatial light modulators (SLMs)[18], chip-based optical vortex emitters offer a much more compact and robust solution. To date, most chip-scale OAM beam emitters are based on silicon or silicon-on-insulator (SOI) wafers[19–22], which are attractive because of the strong optical confinement and mature fabrication technology. However, the indirect bandgap of silicon means that an external laser must be employed to pump the OAM emitter, thus making monolithically integrated OAM emitters very challenging.

Different approaches to implementing monolithic OAM emitters have also been reported. A direct OAM emitter can be realized by fabricating a micro-scale spiral phase plate within the aperture of a vertical-cavity surface-emitting laser[23]. Unfortunately, the device emits light at approximately 860 nm, which is not a telecom wavelength. Optical vortex emission at 1550 nm is demonstrated with a microlaser capable of precise control of the topological charge by taking advantage of the exceptional point formed by complex refractive index modulation[24]. However, the device is optically pumped, which limits its practical applications in telecommunications. A monolithically integrated electrically pumped continuous-wave OAM emitter operating at 1550 nm is highly desirable for OAM-based telecommunications.

In this work, an electrically pumped OAM emitter operating at telecom wavelengths is demonstrated. The device is fabricated on an InGaAsP/InP multiple quantum well (MQW) epitaxial wafer by monolithically integrating a microring-based optical vortex emitter with a distributed feedback (DFB) semiconductor laser. The electrically pumped device, which we shall refer to as an OAM laser hereafter, can emit OAM beam with high-mode purity, making it attractive for applications in fiber communications or quantum information.

## Results

**Device structure.** A microring resonator is likely the most efficient structure for OAM beam generation. For example, an OAM beam is generated and emitted by the same microring structure in a microlaser[24]. However, the symmetry of the ring resonator normally leads to simultaneous generation of clockwise (CW) and counter-CW (CCW) whispering gallery modes (WGMs), and special attention must be paid to avoid mode degeneracy[24].

In this work, a microring-based optical vortex emitter is monolithically integrated with a single-mode laser diode on InP substrate, yielding an electrically pumped OAM laser at telecom wavelengths. As shown in Fig. 1a, light from the DFB laser is coupled into the optical vortex emitter, and an OAM-carrying beam will be emitted vertically once the light is on resonance within the microring. As light is unidirectionally coupled into the

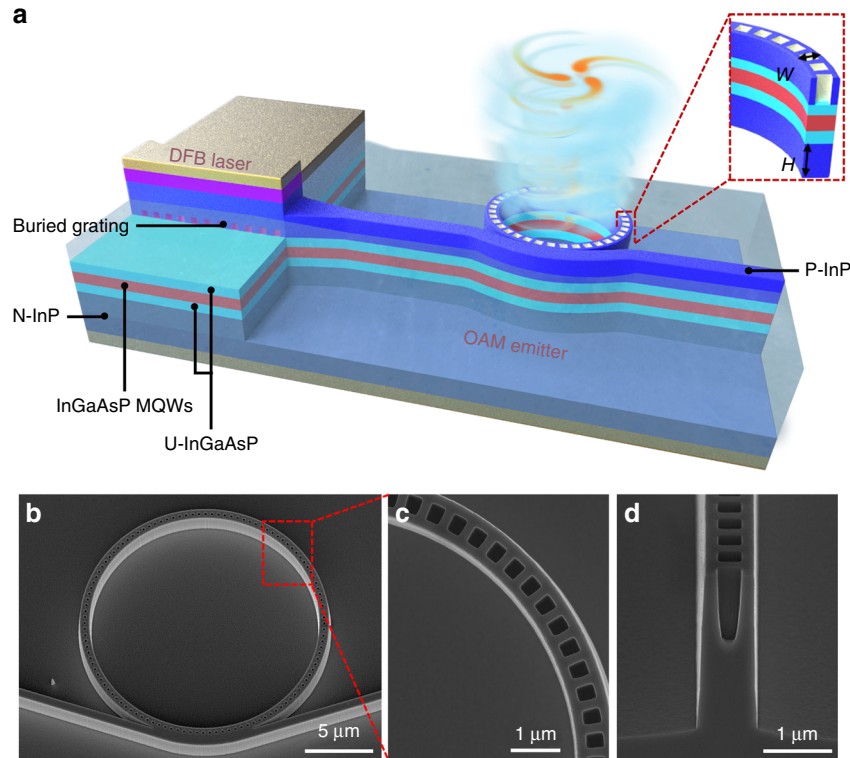

**Fig. 1** Integrated orbital angular momentum laser. **a** Integrated OAM laser with shallow-etched DFB laser and deeply etched vortex emitter on InGaAsP/InP wafer. Inset reveals the cross-section of the vortex emitter. **b** SEM image of the fabricated OAM emitter. **c**, **d** Top and cross-sectional view of the top-grating structure

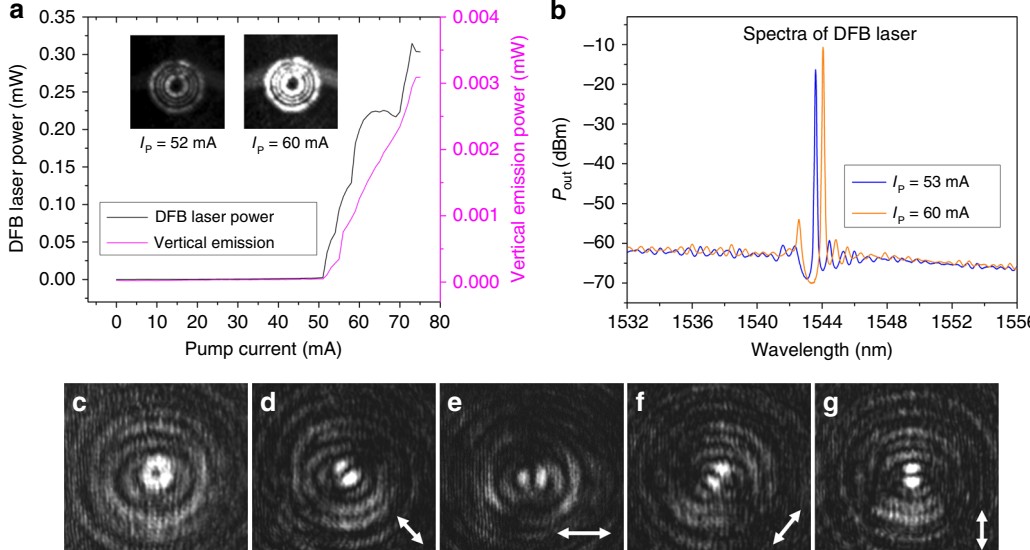

**Fig. 2** *L–I* characteristics and polarization. **a** *L–I* characteristics of DFB laser output power (black) and OAM emission (magenta). Insets show near-field patterns of the integrated OAM laser at different pumping currents $I_P$. **b** Spectrum of DFB laser at different pump current. **c** Intensity distribution of the emitted beam. **d–g** Intensity distributions with the polarization indicated by the arrows

microring, only the CCW mode is excited, thus avoiding mode degeneracy in the WGM resonator.

Compared with OAM emitters based on a circular grating coupler[20] or a microring resonator with download units[21,22], optical vortex emitters based on a microring with angular gratings are highly compact. The radius of the microring can only be a few micrometers long, making the device particularly attractive for dense photonic integration[19]. Furthermore, OAM beams with high-mode purity can be obtained easily by increasing the number of grating elements[25]. In this work, we adopt azimuthal gratings formed on top of the ring waveguide to extract phase-coherent beam from the WGM in the ring. OAM-carrying beams are generated when the following angular phase-matching condition is satisfied[19]

$$l = M - N \qquad (1)$$

where $l$, $M$, and $N$ are all integers, with $l$ being the topological charge of the OAM state, $N$ the number of total grating elements, and $M$ the azimuthal resonant order of the WGM.

As shown in Fig. 1a, the 400 μm-long DFB laser is processed into a shallow-etched ridge waveguide structure, with compressively strained InGaAsP MQWs as the active layer. On the other hand, the 9.56 μm-radius optical vortex emitter adopts a deeply etched ridge structure to ensure strong optical confinement, thus allowing a small ring radius to reduce absorption loss in the ring resonator. To further reduce the absorption loss of the vortex emitter, we adopt the identical epitaxial layer integration scheme with a properly red-detuned lasing wavelength[26]. Thus, the lasing wavelength of the DFB laser lies outside the high absorption range of the InGaAsP MQW active layer, while sufficient gain can be maintained for the lasing mode because of carrier-induced bandgap shrinkage (see Methods). Consequently, both the DFB laser and the vortex emitter can be formed on the same epitaxial layer, without the need for additional regrowth.

The inset of Fig. 1a depicts the geometry of the deeply etched ring waveguide. The mode profile within the ring depends critically on the width and height of the ridge waveguide. A ridge width $W$ of 0.8 μm is adopted to ensure single-mode operation.

Moreover, the distance between the ridge bottom and the lower bound of the InGaAsP cladding layer, denoted by $H$, should be at least 1 μm to ensure negligible optical leakage into the InP substrate (see Supplementary Note 1).

Unlike the case of an SOI-based vortex emitter, in which angular gratings are positioned on the sidewall of the ring to scatter the azimuthal component $E_\varphi$[19], blind-hole-like grating elements measuring $0.4 \times 0.3$ μm² are etched into the top of the ring waveguide to generate vertical OAM radiation by scattering the radial component $E_r$ of the WGM in the InP-based microring, resulting in a radially polarized OAM-carrying beam (see Supplementary Note 1). Figures 1b, c show the scanning electron microscopy images of the OAM laser with top gratings fabricated on the InGaAsP/InP epitaxial wafer. As shown in Fig. 1d, the depth of the top gratings is controlled such that the MQW active layer remains intact. A total of $N = 116$ grating elements are formed on the 9.56 μm-radius WGM resonator, corresponding to a resonance order $M = 120$ at ~1544 nm. Based on the measured transmission and emission spectra (see Supplementary Figs. 7 and 8), the average 3 dB bandwidth of the vortex emitter is estimated to be ~3.15 nm.

**L–I characteristics and polarization.** In Fig. 2a, the optical power collected at the DFB laser facet and vertical to the OAM emitter is plotted against the continuous-wave current injected into the DFB laser (see Supplementary Fig. 9), indicating a threshold current of approximately 51 mA (threshold current density 6375 A cm$^{-2}$). The insets illustrate the near-field patterns of the OAM laser at pump currents near and above the threshold. As illustrated in Fig. 2b, the wavelength of the DFB laser is approximately 1544 nm at an injection current of 60 mA.

Figure 2c shows the far-field intensity distribution of the vertically emitted beam recorded with an infrared charge-coupled device (IR-CCD). The concentric annular pattern with a dark center is characteristic of an OAM-carrying beam[19]. Theoretical analysis predicts radially polarized emission because of the scattering of $E_r$ by top gratings (see Supplementary Notes 1 and 2). To verify the polarization, we measured the intensity distribution after a linear polarizer with the polarization orientation indicated by arrows shown in Figs. 2d–g. It is evident

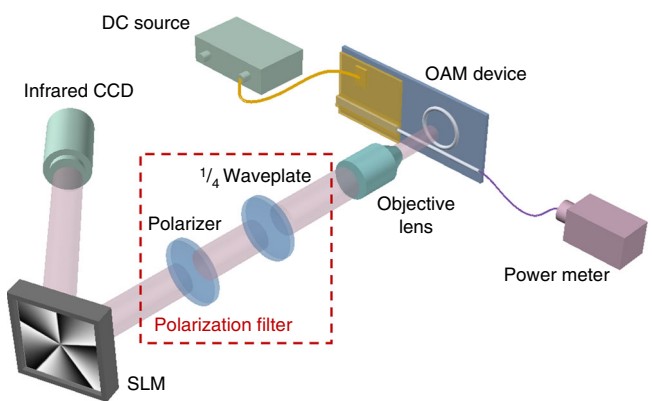

**Fig. 3** Experimental setup for OAM beam characterization. The topological charge and mode purity of the OAM beam is measured with an SLM in combination with a polarization filter

that the two-lobed pattern is parallel to the polarizer axis, confirming radial polarization of the radiated beam.

**OAM mode characterization**. The OAM beam radiated by gratings on the WGM resonator can be expressed in terms of the Jones vector as[25,27]

$$\mathbf{E} = E_{c1} \exp[j(l+1)\varphi] \begin{pmatrix} 1 \\ -j \end{pmatrix} + E_{c2} \exp[j(l-1)\varphi] \begin{pmatrix} 1 \\ j \end{pmatrix}, \quad (2)$$

which is composed of a right-hand circularly polarized (RHCP) OAM beam with topological charge $l+1$ and a left-hand circularly polarized (LHCP) OAM beam with topological charge $l-1$[19,25]. To obtain a pure OAM mode, a polarization filter can be used to select the RHCP or LHCP component of the beam[28].

To verify the topological order and the mode purity, we demodulated the OAM beam by a SLM encoded with different orders of helical phase[28,29]. Figure 3 shows the setup used to determine the topological charge of the OAM beam. The OAM laser is pumped by a direct current source and the OAM-carrying beam is collimated by an objective lens. The LHCP or RHCP component of the beam is selected when the 1/4 waveplate is set to $+45°$ or $-45°$ relative to the axis of the polarizer. The beam is then demodulated and reflected by the SLM and finally projected onto an IR-CCD. The topological charge of the OAM-carrying beam can be determined by varying the order of the holographic pattern loaded onto the SLM.

As shown in the first row of Fig. 4a, the OAM order of the RHCP component is found to be $+5$, as a bright spot at the center appears when the beam is reflected by the SLM with a $-5$-order holographic phase pattern. On the other hand, the OAM order of the LHCP component is $+3$. According to equation (2), the $l$ value of this OAM laser is $+4$.

With the help of the SLM, the mode purity of the emitted OAM beam can also be measured. Here, the OAM mode purity is defined as the relative central intensity of the beam after demodulation by the SLM with opposite OAM order[28,29]. As illustrated in Fig. 4b, the mode purity values of the RHCP and LHCP components are determined to be 0.76 and 0.85, respectively. For our integrated OAM laser, the degradation of mode purity may result from the mismatch between the lasing wavelength and the WGM resonator resonance. Wavelength matching could be fulfilled by either tuning the wavelength of the DFB laser or adjusting the effective refractive index of the WGM resonator by microheaters[28].

To determine the ideal mode purity attainable with our OAM laser, i.e., mode purity under wavelength matching, we send the

light from a tuneable laser into the OAM emitter without electrically pumping the laser and demodulate the vertically emitted beam with an SLM. Figs 4c, d plot the OAM mode purity of different OAM orders obtained by adjusting the incident laser wavelength to coincide with that of the corresponding resonance. The histograms indicate that for our OAM emitter, a mode purity higher than 0.8 for most modes could be realized upon wavelength matching. Furthermore, clear helical phase structure can be captured via interference with a reference beam (see Supplementary Figs. 10 and 11).

The inhomogeneous light intensity in the WGM resonator is another important factor leading to degraded OAM mode purity[25]. As light propagates inside the WGM resonator, the intensity of the light will gradually decrease because of various loss mechanisms, including the scattering of the grating and absorption of the III–V material. This degradation can be alleviated by optimizing the coupling structure, adjusting the geometry of the grating elements[30], and electrically pumping the WGM resonator.

## Discussion

An electrically pumped integrated OAM laser at telecom wavelengths is successfully demonstrated. The DFB laser integrated optical vortex emitter allows unidirectional WGM excitation in the microring resonator, thus effectively avoiding mode degeneracy. The optical vortex emitter is equipped with a unique top-grating structure, which enables efficient OAM radiation in the vertical direction. As a result, OAM lasing with specific topological charge is demonstrated. In addition, the InP-based integrated OAM laser is highly compact, as the size of the deeply etched WGM resonator can be reduced to the micrometer level.

The integrated OAM laser eliminates the external laser required by silicon- or SOI-based vortex emitters and thus opens a path toward large-scale integration. Furthermore, the topological charge of the OAM beam can be tuned either by adjusting the wavelength of the DFB laser or by thermally tuning the microring, which would greatly broaden its applications.

When the integrated OAM laser is to be used for free-space or fiber communications, high-bit-rate data can be imprinted on the OAM beam by direct modulation of the DFB laser or by an integrated optical modulator. One of the OAM components (e.g., the RHCP component) with imprinted data will be selected by a polarization filter (like the one used in Fig. 3) and will subsequently be transmitted through free-space or fed into a fiber.

Furthermore, the vortex beams generated by the integrated OAM lasers are actually vector vortex beams[31], which are not only interesting for imaging or optical trapping, but can also form an orthogonal modal basis set for information encoding or multiplexing. It has already been demonstrated that such vector vortex beams can be employed for data transmission through free-space[32] or via optical fibers[33], suggesting the potential of the integrated OAM laser in SDM communication systems, in which the power of the vortex beam emitted by the device can be fully exploited.

## Methods

**Device fabrication**. During device fabrication, buried gratings are formed in the DFB laser section through two-step epitaxial growth[26]. The grating pitch is judiciously chosen to ensure that the Bragg wavelength of the DFB laser is redshifted from the photoluminescence peak of the compressively strained InGaAsP MQW active layer by approximately 45 nm. The DFB laser section is then processed into a standard ridge waveguide structure, with the width and height of the ridge being 2 and 1.95 μm, respectively. As depicted in Fig. 1a, the ohmic contact layer and part of the top InP cladding layer over the vortex emitter section are removed to reduce the absorption and scattering loss of the top gratings. The height of the deeply etched vortex emitter is approximately 2.1 μm.

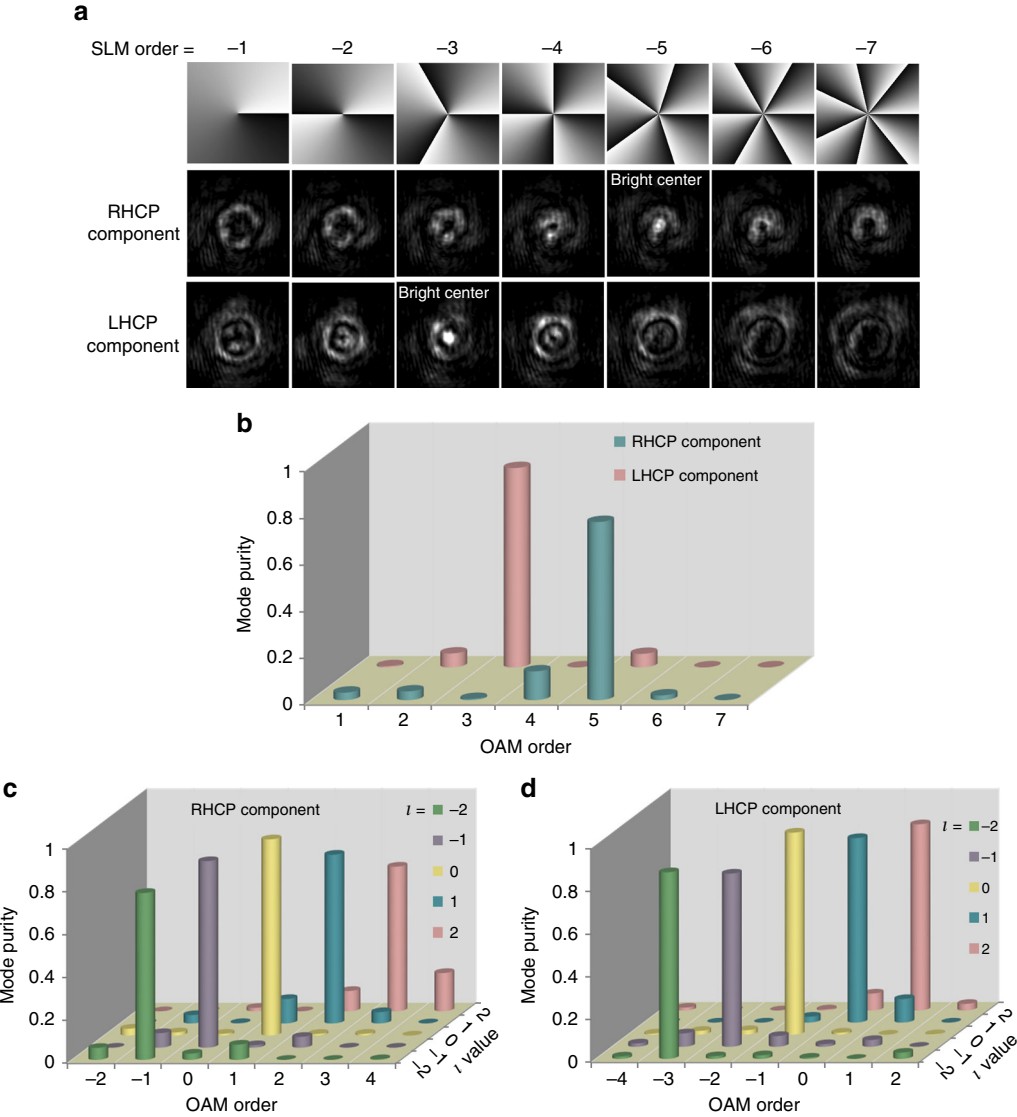

**Fig. 4** OAM mode characterization. **a** OAM lasing modulated by SLM. The first row shows holograms of the SLM. The second and third rows are patterns of the RHCP and LHCP components after being reflected by the SLM in the first row, indicating $l = +4$. **b** Mode purities of the OAM lasing measured by the SLM. **c** Mode purities of the RHCP and **d** LHCP components of the OAM-carrying beams radiated by the OAM emitter

As shown in Fig. 1b, a pulley coupler is adopted to increase the coupling length and ensure sufficient coupling between the ring and the bus waveguide, which are separated by a gap of 50 nm. To ensure precise control of the mask pattern close to the coupling section, different exposure doses are employed for the WGM resonator near and far from the pulley coupling structure to alleviate the proximity effect of electron beam lithography.

The deeply etched waveguide with grating holes is formed by a single-step inductively coupled plasma (ICP) dry-etching process using a mixture of $CH_4$ and $H_2$. This process is made possible by taking advantage of the lag effect, i.e., the etch rate is dependent on the mask opening. Thereby, the etch rate within the grating holes is lower than in the open area. By adjusting the size of the grating holes and tuning the dry-etching conditions, e.g., the gas ratio or the ICP power, precise control of the etching depth can be guaranteed (see Supplementary Note 3). As a result, the MQW layer beneath the grating region remains intact, thus ensuring low transmission loss in the WGM resonator.

**Data availability**. The authors declare that the main data supporting the findings of this study are available within the article and its Supplementary Information. Extra data are available from the corresponding author upon reasonable request.

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

## Acknowledgements

This work was supported in part by National Basic Research Program of China (2014CB340002); National Natural Science Foundation of China (51561165012, 61574082, 61621064, and 11690031); Tsinghua University Initiative Scientific Research Program (20161080068, 20161080062).

## Author contributions

J.Z. and C.S. designed the devices. J.Z. carried out the fabrication. Y.X., C.Y., T.T., and Y. N. contributed to the fabrication. J.Z., C.S., S.L., and X.C. performed the measurements. All of the authors contributed to the data analysis and authorship of the manuscript. C.S. and S.Y. developed and supervised the project.

## Additional information

**Competing interests:** The authors declare no competing interests.

