## [Peer Review File · Nature Communications]

Reviewers' comments:

Reviewer #1 (Remarks to the Author):

This paper discusses the first, according to the authors, demonstration of a monolithically integrated laser structure that emits OAM beams. The idea combines a conventional DFB laser with a resonator based OAM emitter similar in operation principle to one demonstrated by a subset of the same authors in their Science paper in 2012 (ref. 19 in this manuscript). Although achieved mode purities (0.76 to 0.85) require optimisation for practical applications, the authors provide convincing assertions on pathways to improve device quality. Overall, this is a nice demonstration, although this reviewer would like to see some discussion that justifies this approach further, in comparison to existing approaches that can produce very high purity OAM beams:

(1) While the demonstration of a monolithically integrated device is indeed interesting, it is not clear this is a better approach than hybrid integration, where a DFB laser is incorporated on a Si or SOI chip – indeed, for several other optical functionalities currently envisaged with waveguide devices, this is the preferred approach considering that InP real estate is significantly more costly than Si or SOI real estate.

(2) Perhaps related to the above comment, why is their efficiency so low? From Fig. 2a, one infers that ~1% of the light from the DFB laser is finally emitted by the resonator in an OAM beam. Is this inherent to the OAM emitter's architecture, or were there other losses connecting the DFB laser section with the resonator? If the latter, doesn't that defeat the purpose of monolithic integration?

(3) Other concerns regarding device performance relate to the characteristics of the emitter itself, and the queries below may concern the emitter demonstrated here as well as the original similar emitter the authors described in their Science paper:

(a) As with the original demonstration (and as discussed here too, in the context of getting the wavelength right), this emitter works best only at a specific wavelength (or wavelengths separated by an appropriate free-spectral range). That should not be a problem considering that this is an integrated emitter designed to work with a specific DFB laser on the same chip (so the two can be tuned together, as the authors have done here). However, this wavelength selectivity also brings up the question of how narrowband this emitter is. Is it so narrowband that one needs to use an external modulator if one were to encode data (which would necessarily broaden the DFB emission's linewidth)? In this reviewer's opinion, a discussion on this bandwidth limitation would be important for a paper that discusses such a demonstration.

(b) The emission is radially polarised, which when broken into circular polarisation components, actually yields two distinct OAM beams. As such, how do the authors envisage using this device? For free-space communications, are they aiming to encode data on the radially polarised beams directly emitted from the emitter? In fiber communications, are they intending to separate the two polarisations, modulate them, and then feed them into fibers?

The practical consideration arising from these (3a, 3b) concerns is that using this device requires additional external devices, as opposed to direct modulation of the DFB, which would represent a truly compact device. This doesn't mean the device is not interesting, but it would be useful for the authors to discuss these and other pros and cons of the fundamental constraints of their device architecture.

Reviewer #2 (Remarks to the Author):

The manuscript "Electrically pumped orbital angular momentum (OAM) laser at telecom wavelengths" proposes and demonstrates a laser monolithically integrated with a ring resonator containing a grating that emits a linear combination of orbital angular momentum modes. While this is an ambitious endeavor and the device successfully works, I do not believe the result is significant enough for publication in Nature Communications. There are no new ideas presented, the output power is very low, and the usefulness is difficult to discern. My detailed comments are as follows:

1) The concept of producing OAM modes from a ring resonator containing a grating is not new. It was already presented in Science in Ref. 19. Thus the main new parts are integrating this with a laser and making the resonator in InP instead of Si. However, it does not appear advantageous to make these changes.

2) A significant difficulty in making a surface-emitting grating in InP is that it is not clear that the waveguide with the grating can have an effective index that is higher than the lower cladding. The authors make a very deep etch, but this may not be enough. Leakage into the substrate could be one of the reasons the output power is so low. Does the mode of Fig. 5b include the grating?

3) The authors make multiple incorrect claims in the Introduction. They say "OAM offers a new degree of freedom...". This is not true. OAM is just a linear combination of other orthonormal basis sets: polarization and space. In fact, it has been shown that OAM is an inferior way to do space

division multiplexing that to use conventional orthogonal spatial modes. The authors say that it is impossible to make a monolithically integrated OAM emitter in Si. This could be done by using Er-doped waveguides on silicon.

4) The title is misleading. The title makes it sounds like this is a paper about a device that emits an OAM directly from a laser. In reality, the device is a laser which feeds a grating; the OAM emitter is independent of the laser.

5) More detail on the DFB laser should be given. It does not appear to be a quarter-wave-shifted DFB. In such a case, the exact lasing wavelength is difficult to control. How is it aligned to the grating in the ring?

6) The output power is very low. The authors need to comment on this. With such a low output power, it is difficult to understand how this device could be used.

7) The device outputs a linear combination of two OAM modes. The usefulness of this should be discussed. It seems to defeat the purpose of using OAM modes for multiplexing. How would data be imprinted on the OAM modes?

Reviewer #3 (Remarks to the Author):

An electrically pumped OAM laser operating at telecom wavelengths was designed and fabricated by 20 monolithically integrating an optical vortex emitter with a distributed feedback (DFB) laser on the same InGaAsP/InP epitaxial wafer.

This is an important and innovative idea and a novel device concept that I believe would be of interest for the wide community of researchers.

I recommend this paper for publication after the following points are addressed:

1. The title is a bit misleading as what is really demonstrated is integrated DFB laser with a vortex emitter and not strictly speaking electrically pumped OAM laser. This does not make this device less important, but perhaps more straightforward title could be considered. In a sense, vortex emitted coupled with an external laser source have been previously demonstrated, so I believe the novelty of the device developed in this paper is in integration.
2. It would be helpful to explain the shape (oscillations) of the L-I characteristics of DFB laser output.
3. Spectral response at different current levels is of interest to understand the mode content of the device.

Response to Reviewers' comments

Reviewer #1

(1) *While the demonstration of a monolithically integrated device is indeed interesting, it is not clear this is a better approach than hybrid integration, where a DFB laser is incorporated on a Si or SOI chip – indeed, for several other optical functionalities currently envisaged with waveguide devices, this is the preferred approach considering that InP real estate is significantly more costly than Si or SOI real estate.*

Answer: CMOS compatible silicon photonics is certainly very attractive for its potential in implementing large-scale photonic integration circuits (PICs) and reducing device cost. Impressive results have been demonstrated by incorporating III-V material-based laser sources into silicon photonics via hybrid integration. Nevertheless, such approach still faces some challenges.

1. Firstly, hybrid lasers formed by wafer bonding technique still suffer from relatively low output power as well as a limited operating temperature.
2. Wafer bonding process not only makes the fabrication of hybrid lasers complicated, but may also raise reliability concerns. There have been experimental evidences showing that the optical and electrical characteristics of hybrid laser diodes fabricated by low temperature molecular wafer bonding technique may degrade due to diffusion of non-radiative defects [R1].

On the other hand, InP is a very versatile platform, allowing the fabrication of both high performance active components such as laser diodes, optical amplifiers, modulators and photodetectors, and passive components such as optical filters, splitters and multiplexers. Monolithic integration of different components on the same InP wafer offers the benefits of reduced power consumption and improved reliability. With the advent of InP-based generic integration technology [R2], it is expected that the cost of InP-based PICs will be significantly reduced. As a result, we consider that the monolithically integrated OAM laser demonstrated in this work has promising potential in practical applications.

[R1] M. Buffolo, *et al.* “Degradation mechanisms of heterogeneous III-V/silicon 1.55- μm DBR laser diodes,” *IEEE J. Quantum Electron.* **53**, 8400108 (2017).

[R2] M. Smit, *et al.*, “An introduction to InP-based generic integration technology.” *Semicond. Sci. Technol.* **29**, 083001 (2014).

(2) Perhaps related to the above comment, why is their efficiency so low? From Fig. 2a, one infers that ~1% of the light from the DFB laser is finally emitted by the resonator in an OAM beam. Is this inherent to the OAM emitter’s architecture, or were there other losses connecting the DFB laser section with the resonator? If the latter, doesn’t that defeat the purpose of monolithic integration?

Answer: The relatively low efficiency of our prototype device may be caused by the following issues:

1. Insufficient coupling between the bus waveguide and the microring: In our OAM laser, a deeply-etched ridge structure is adopted for the vortex emitter to reduce the waveguide bending loss. Strong optical confinement of the deeply-etched waveguides makes it difficult to realize critical coupling between the bus waveguide and the microring, as evident from the transmission spectrum shown in Supplementary Fig. 2. An enhanced coupling between the bus waveguide and microring can be achieved by optimizing the coupling structure.
2. Half of the emitted power wasted as downward emission: In addition to the vortex beam emitted upward and detected by the CCD camera, the top gratings on the microring induce a downward radiation, thus halving the detectable emission efficiency. A possible remedy to this problem is to block the downward emission by a distributed Bragg reflector (DBR) mirror judiciously positioned beneath the waveguide core.
3. Non-ideal wavelength matching between the DFB laser and the vortex emitter. Wavelength matching not only influences the mode purity of the emitted OAM beam, but also limits the emission efficiency. Improved wavelength matching between the lasing wavelength of the DFB laser and the resonance of the microring can be realized by tuning the effective refractive index of the WGM resonator by microheaters
4. Residual absorption in the vortex emitter section. In this proof-of-concept demonstration, we have adopted the identical epitaxial layer (IEL) integration scheme to simplify the fabrication of the OAM laser. Both the DFB laser and the vortex emitter share the same InGaAsP MQW active layer, and the lasing wavelength is redshifted from the luminescence peak of the active layer by about 45 nm. The residual absorption in the bus waveguide and the microring would lead to a reduced emission efficiency. This problem can be solved by electrically pumping the microring, or by adopting the quantum-well intermixing (QWI) integration scheme to enhance the bandgap of the vortex emitter section.

Apart from the points discussed above, further optimization of the device structure would be necessary to enhance the emission efficiency, e. g. the transition from the shallow-etched waveguide in the DFB laser section to the deeply-etched waveguide in the vortex emitter section. We believe that by optimizing the device structure as well as the fabrication procedure, emission efficiency comparable to or even better than that demonstrated with silicon-based vortex emitters (up to 13%) is attainable for the integrated OAM laser.

(3) Other concerns regarding device performance relate to the characteristics of the emitter itself, and the queries below may concern the emitter demonstrated here as well as the original similar emitter the authors described in their Science paper:

(a) As with the original demonstration (and as discussed here too, in the context of getting the wavelength right), this emitter works best only at a specific wavelength (or wavelengths separated by an appropriate free-spectral range). That should not be a problem considering that this is an integrated emitter designed to work with a specific DFB laser on the same chip (so the two can be tuned together, as the authors have done here). However, this wavelength selectivity also brings up the question of how narrowband this emitter is. Is it so narrowband that one needs to use an external modulator if one were to encode data (which would necessarily broaden the DFB emission's linewidth)? In this reviewer's opinion, a discussion on this bandwidth limitation would be important for a paper that discusses such a demonstration.

Answer: Unlike microring based filters, where a high quality factor (Q -factor) is often mandatory, the microring-based vortex emitter exhibits only a moderate Q -factor, since a considerable part of the optical power coupled into the microring is lost in the form of vertical emission of vortex beam. Consequently, the vortex emitter (either the original one reported in the Science paper or the integrated one demonstrated here) is not so narrowband.

According to the transmission and emission spectra illustrated in Supplementary Fig. 2, the average 3-dB bandwidth of emission peaks in our vortex emitter is ~ 3.15 nm, which is wide enough to accommodate light encoded with high-bit-rate data up to 33 GBaud, the standard for current high-capacity coherent optical transmissions. In practical applications, high-bit-rate data can be encoded by direct modulation of the DFB laser, or by a monolithically integrated optical modulator, e.g. an electroabsorption modulator (EAM) or a Mach-Zehnder modulator (MZM) inserted between the DFB laser and the vortex emitter.

(b) The emission is radially polarised, which when broken into circular polarisation components, actually yields two distinct OAM beams. As such, how do the authors envisage using this device? For free-space communications, are they aiming to encode data on the radially polarised beams directly emitted from the emitter? In fiber communications, are they intending to separate the two polarisations, modulate them, and then feed them into fibers?

Answer: When the integrated OAM laser is to be used for free-space or fibre communications, high-bit-rate data can be encoded by direct modulation of the DFB laser or by an integrated optical modulator (e.g. a monolithically integrated EAM). One OAM component (e.g. the RHCP component) of the data encoded vortex beam will be selected by a polarization filter (like the one used in Fig. 3), and subsequently transmitted through free-space or fed into a fibre.

The above approach has the advantage of being relatively simple, though it suffers from reduced efficiency, as half of the power in the vortex beam (i.e. the LHCP component) has to be discarded. To make full use of both RHCP and LHCP components would require modulating the OAM beams after separating them from each other. As so far there is no integrated device capable of high-speed modulation of OAM-carrying beams, this approach is not considered preferable. Nevertheless, the two OAM components encoded with the same data can be employed for multicasting of high-bit-rate data.

In addition, the beams generated from our device are actually vector vortex beams. These vector vortex beams can also be considered as another orthogonal modal basis set for information encoding or multiplexing. It has already been demonstrated that such vector vortex beams can be employed for data transmission through free-space [R3] or via optical fibres [R4]. This means that the vector vortex beams have the potential to be employed in SDM communication systems, and

the vortex beam generated from our devices can be fully exploited without any loss.

We have included a paragraph in the Discussion of revised manuscript, so as to clarify how the integrated OAM laser is to be used for free-space or fibre communications.

[R3] G. Milione, *et al.*, “4×20 Gbit/s mode division multiplexing over free space using vector modes and a q-plate mode (de)multiplexer,” *Opt. Lett.* **40**, 1980-1983 (2015).

[R4] J. Liu, *et al.*, “Direct fiber vector eigenmode multiplexing transmission seeded by integrated optical vortex emitters,” *Light Sci. Appl.* **7**, 17148 (2018).

The practical consideration arising from these (3a, 3b) concerns is that using this device requires additional external devices, as opposed to direct modulation of the DFB, which would represent a truly compact device. This doesn't mean the device is not interesting, but it would be useful for the authors to discuss these and other pros and cons of the fundamental constraints of their device architecture.

Answer: As discussed in our reply to 3a and 3b, the integrated OAM laser does not require external devices for data encoding, which can be implemented either by direct modulation of the DFB laser, or external modulation via an optical modulator monolithically integrated on the same wafer. So the integrated OAM laser is a truly compact device. On the other hand, separating the preferred OAM component (RHCP or LHCP) and multiplexing beams with distinct OAM orders require external devices. We are still trying to figure out the possible way to implement such functionality onto the same chip.

Reviewer #2:

1) *The concept of producing OAM modes from a ring resonator containing a grating is not new. It was already presented in Science in Ref. 19. Thus the main new parts are integrating this with a laser and making the resonator in InP instead of Si. However, it does not appear advantageous to make these changes.*

Answer: An InP-based monolithically integrated OAM laser is very attractive, as it provides a very compact device for OAM beam generation, and will be valuable for fibre communications, microscopic particle manipulation, and quantum applications.

Though microring based vortex emitters have already been demonstrated in silicon, monolithic integration of such a vortex emitter with a laser on InP substrate proves nontrivial. For the vortex emitter to work on InP-based epitaxial wafer, radically different grating design has to be employed, and special fabrication technique has been developed accordingly.

Furthermore, high-bit-rate data encoding can be realized by direction modulation of the DFB laser or by an optical modulator (e.g. an electroabsorption modulator or a Mach-Zehnder modulator) monolithically integrated on the same InP wafer, thus forming a high-speed OAM signal generator.

2) *A significant difficulty in making a surface-emitting grating in InP is that it is not clear that the waveguide with the grating can have an effective index that is higher than the lower cladding. The authors make a very deep etch, but this may not be enough. Leakage into the substrate could be one of the reasons the output power is so low. Does the mode of Fig. 5b include the grating?*

Answer: As correctly pointed out by the Reviewer, our simulations show that a narrow ridge waveguide may lead to significant mode leakage into the lower InP cladding. However, by increasing the depth of the deeply etched ridge, we can ensure negligible optical leakage, as illustrated in Figs. 5(a) and 5(b).

Figure R1 shows the electric field distribution with and without gratings on the deeply etched straight waveguide. By comparing Fig. R1(b) with Fig. R1(d), it is evident that the introduction of top gratings will not bring about significant impacts on the light leakage, as the depth of the top gratings are so controlled as to leave the MQW layer intact.

In our prototype device, a narrow ridge width of $0.8\ \mu\text{m}$ is adopted to ensure single mode operation for the microring, thus avoiding any possible ambiguity in WGM extraction. In principle, a wider ridge can be employed, which would be helpful in further reducing the optical leakage into substrate. As different modes in the microring will exhibit different free spectrum ranges, it is possible to selectively excite the preferred mode by aligning the lasing wavelength of the DFB laser to the specific resonance, thus ensuring high OAM mode purity even for a multi-mode microring.

Figure R1 | Electric field in the deeply etched straight waveguide. (a) Azimuthal electric field component E_ϕ and (b) transverse electric field component E_r for the quasi-TE mode in deeply etched straight waveguide with no grating, and the wavelength is 1550 nm. (c) E_ϕ and (d) E_r in waveguide with gratings at the sidewall and on the top respectively. The wavelength is 1550 nm.

3) The authors make multiple incorrect claims in the Introduction. They say "OAM offers a new degree of freedom...". This is not true. OAM is just a linear combination of other orthonormal basis sets: polarization and space. In fact, it has been shown that OAM is an inferior way to do space division multiplexing that to use conventional orthogonal spatial modes. The authors say that it is impossible to make a monolithically integrated OAM emitter in Si. This could be done by using Er-doped waveguides on silicon.

Answer: We agree with the Reviewer that OAM is actually a special form of space-division multiplexing (SDM), which employs a set of orthonormal bases with circular symmetry. Accordingly, we have revised the part in the Introduction as follows: As a special form of space-division multiplexing (SDM), OAM offers an extra degree of freedom for information encoding, in addition to the already extensively exploited wavelength, amplitude, phase, and polarization.

There has been some controversy concerning the capacity limits of OAM, particularly in free-space transmission of RF or acoustic waves [R5, R6]. Nevertheless, OAM based SDM has been proved attractive in fibre communications. Terabit-scale data transmission has been demonstrated by employing OAM in specially designed vortex fibres [R7]. Low-complexity digital-signal-processing (DSP) coherent detection methods can be used, without resorting to

computationally intensive DSP corrective algorithms, such as multiple-input-multiple-output (MIMO). Recently, 8 OAM modes multiplexing transmission over 10 km ring core fibers has been demonstrated, and an aggregate capacity of 5.12 Tb/s with an overall spectral efficiency of 9 bit/s/Hz is realized [R8]. Thanks to the low cross-talk between OAM mode groups, only 4×4 MIMO processing is required, illustrating desirable trade-off between system capacity and DSP complexity. Monolithically integrated OAM lasers are attractive for such communication systems and will play a crucial part in exploring their capacity limits.

Furthermore, in addition to fibre communications, OAM lasers presented in this work would be of interest for microscopic particle manipulation, quantum key distribution or high-dimensional quantum cryptography.

Er-doped waveguides have been employed to form amplifiers monolithically integrated with silicon waveguides. However, external lasers are required to act as pump to provide gain for light amplification or lasing. On the other hand, quantum dot lasers formed by direct growth of III-V materials on silicon substrates have demonstrated promising results, though the technique still faces considerable challenges. As a result, we have revised the part in the Introduction as follows: However, the indirect bandgap of silicon means that an external laser must be employed to pump the OAM emitter, thus making monolithically integrated OAM emitters very challenging.

[R5] N. Zhao, *et al.*, “Capacity limits of spatially multiplexed free-space communication,” *Nat. Photonics* **9**, 822–826 (2015).

[R6] D. A. B. Miller, “Better choices than optical angular momentum multiplexing for communications,” *PNAS* **114**, E9755–E9756, (2017).

[R7] N. Bozinovic, *et al.*, “Terabit-scale orbital angular momentum mode division multiplexing in fibers,” *Science* **340**, 1545–1548 (2013).

[R8] G. Zhu, *et al.*, “Scalable mode division multiplexed transmission over a 10-km ring-core fiber using high-order orbital angular momentum modes,” *Opt. Express* **26**, 594–604 (2018).

4) *The title is misleading. The title makes it sounds like this is a paper about a device that emits an OAM directly from a laser. In reality, the device is a laser which feeds a grating; the OAM emitter is independent of the laser.*

Answer: To better summarize our work, we have revised the title of our manuscript to “Monolithically integrated orbital angular momentum (OAM) laser at telecom wavelengths.”

5) *More detail on the DFB laser should be given. It does not appear to be a quarter-wave-shifted DFB. In such a case, the exact lasing wavelength is difficult to control. How is it aligned to the grating in the ring?*

Answer: Buried gratings with uniform grating pitch are formed in the DFB laser section by holographic exposure, chemical etching and subsequent regrowth. As a result, the laser is not a quarter-wave-shifted DFB laser and the lasing wavelength is not precisely located at the Bragg wavelength, but on either side of the stopband.

We take the following procedures to align the wavelength of the DFB laser to the resonance of the microring. Firstly, we measure the lasing wavelength of DFB lasers fabricated on a control wafer taken from the same epitaxial wafer for OAM laser fabrication. Then OAM laser chips with slightly different microring radii and top grating pitches are fabricated. Chips with lasing wavelength aligned to the microring resonance are then selected for further tests.

As discussed in the paper, the alignment between the lasing wavelength and the microring resonance has a crucial impact on the mode purity of the OAM laser. To improve the wavelength alignment, quart-wave-shifted DFB laser can be employed so that the lasing wavelength locates precisely at the Bragg wavelength. In addition, on-chip microheaters will be formed to fine tune the resonance of the microring.

6) *The output power is very low. The authors need to comment on this. With such a low output power, it is difficult to understand how this device could be used.*

Answer: The relatively low efficiency of our prototype device is attributed to the following issues:

1. Insufficient coupling between the bus waveguide and the microring: In our OAM laser, a deeply-etched ridge structure is adopted for the vortex emitter to reduce the waveguide bending loss. Strong optical confinement of the deeply-etched waveguides makes it difficult to realize critical coupling between the bus waveguide and the microring, as evident from the transmission spectrum shown in Supplementary Fig. 2. An enhanced coupling between the bus waveguide and microring can be achieved by optimizing the coupling structure.
2. Half of the emitted power wasted as downward emission: In addition to the vortex beam emitted upward and detected by the CCD camera, the top gratings on the microring induce a downward radiation, thus halving the detectable emission efficiency. A possible remedy to this problem is to block the downward emission by a distributed Bragg reflector (DBR) mirror judiciously positioned beneath the waveguide core.
3. Non-ideal wavelength matching between the DFB laser and the vortex emitter. Wavelength matching not only influences the mode purity of the emitted OAM beam, but also limits the emission efficiency. Improved wavelength matching between the lasing wavelength of the DFB laser and the resonance of the microring can be realized by tuning the effective refractive index of the WGM resonator by microheaters
4. Residual absorption in the vortex emitter section. In this proof-of-concept demonstration, we have adopted the IEL integration scheme to simplify the fabrication of the OAM laser. Both the DFB laser and the vortex emitter share the same InGaAsP MQW active layer, and the lasing wavelength is redshifted from the luminescence peak of the active layer by about 45 nm. The residual absorption in the bus waveguide and the microring would lead to a reduced emission efficiency. This problem can be solved by electrically pumping the microring, or by adopting the quantum-well intermixing (QWI) integration scheme [R9] to enhance the bandgap of the vortex emitter section.

Apart from the points discussed above, further optimization of the device structure would be necessary to enhance the emission efficiency, e.g. the transition from the shallow-etched waveguide in the DFB laser section to the deeply-etched waveguide in the vortex emitter section. We believe that by optimizing the device structure as well as the fabrication procedure, emission efficiency comparable to or even better than that demonstrated with silicon-based vortex emitters (up to 13%) is attainable for the integrated OAM laser.

[R9] H. Huang, *et al.*, “Mode division multiplexing using an orbital angular momentum mode sorter and MIMO-DSP over a graded-index few-mode optical fibre,” *Sci. Rep.* **5**, 14931, (2015).

7) *The device outputs a linear combination of two OAM modes. The usefulness of this should be discussed. It seems to defeat the purpose of using OAM modes for multiplexing. How would data be imprinted on the OAM modes?*

Answer: When the integrated OAM laser is to be used for free-space or fibre communications, high-bit-rate data can be imprinted on the OAM beam by direct modulation of the DFB laser or by an integrated optical modulator. Only one OAM mode (e.g. the RHCP component) with imprinted data will be selected by a polarization filter (like the one used in Fig. 3), and will be subsequently transmitted through free-space or fed into fibre.

In addition, the beams generated from our device are actually vector vortex beams. These vector vortex beams can also be considered as another orthogonal modal basis set for information encoding or multiplexing. It has already been demonstrated that such vector vortex beams can be employed for data transmission through free-space [R3] or via optical fibres [R4]. This means that the vector vortex beams have the potential to be employed in SDM communication systems, and the vortex beam generated from our devices can be fully exploited without any loss.

We have included a paragraph in the Discussion of revised manuscript, so as to clarify how the integrated OAM laser is to be used for free-space or fibre communications.

Reviewer #3:

1. *The title is a bit misleading as what is really demonstrated is integrated DFB laser with a vortex emitter and not strictly speaking electrically pumped OAM laser. This does not make this device less important, but perhaps more straightforward title could be considered. In a sense, vortex emitted coupled with an external laser source have been previously demonstrated, so I believe the novelty of the device developed in this paper is in integration.*

Answer: Following the Reviewer's suggestion, we have revised the title of our paper to "Integrated orbital angular momentum (OAM) laser at telecom wavelengths," so as to better convey the novelty of this work.

2. *It would be helpful to explain the shape (oscillations) of the L-I characteristics of DFB laser output.*

Answer: A series of kinks can be observed in the L-I curve of the DFB laser, as indicated by the arrows in Fig. R2. Such periodic kinks in the L-I characteristics of the DFB laser is attributed to optical feedback due to facet reflection. As both facets of the device chip are left as cleaved, DFB laser output coupled into the bus waveguide will be partially reflected by the facet of the vortex emitter section. As the injection current into the DFB laser section increases, the lasing wavelength will shift to the longer wavelength, thus causing the feedback phase to vary accordingly. This leads to the observed oscillation, i.e. kinks, as indicated by arrows in Fig. R2(a).

The pronounced dip for current around 60 ~ 70 mA is considered to be a result of alignment between lasing wavelength and the microring resonance, i.e. wavelength matching. Such alignment allows a considerable portion of the light fed into the waveguide to be coupled into the microring resonator and converted to vertical radiation. As a result, the optical power reflected into the DFB laser will be significantly reduced. This will cause an increase in the threshold current of the DFB laser, hence the dip in output power collected at the DFB laser facet. The

above analysis is corroborated by the fact that the vertical emission power exhibits no dip in the current range 60-70 mA, since the vertical radiation is enhanced due to wavelength matching.

Such kinks can be suppressed by applying an antireflection coating to the output facet, or by tilting the bus waveguide around the facet.

Figure R2 | L-I characteristics and polarization. (a) L-I characteristics of DFB laser output power (black) and OAM emission (magenta). (b) Spectra of DFB laser at different pump current I_{pump} . (c) OAM lasing modulated by SLM when the lasing wavelength deviates from the microring resonance. The first row shows holograms of the SLM. The second and third rows are patterns of the RHCP and LHCP components after being reflected by the SLM in the first row.

3. Spectral response at different current levels is of interest to understand the mode content of the device.

Answer: Figure R2(b) plots lasing spectra of the DFB laser recorded at different current level. It is evident that the lasing wavelength shifts with the current. Wavelength matching is roughly satisfied at an injection current of ~ 60 mA. As the injection current varies, the lasing wavelength will deviate from the microring resonance, thus leading to a degradation of the OAM mode purity. Figure R2(c) shows the demodulated OAM order when the lasing wavelength deviates from the microring resonance. It is seen that both +5 and +4-order in the RHCP component exhibit a bright spot at the center, indicating degraded mode purity.

REVIEWERS' COMMENTS:

Reviewer #1 (Remarks to the Author):

The authors have done an excellent job of addressing all the comments and concerns of this reviewer. They provide a convincing explanation of the utility of their device in OAM communications. They also provide convincing explanations for the origin of the high losses in their current device. In summary, the device implementation is interesting, the utility of the device is potentially high. However, conceptually it is the combination of two known devices - so whether a first attempt with high losses, such as this one, but with promising and convincing routes for device performance optimisation, suffices for publication in NatComm... or if the authors should come back with a device with far improved (loss) performance... is a decision left to the editor - this reviewer is happy with either decision.

Reviewer #2 (Remarks to the Author):

The authors have addressed all of the Reviewers' comments. I am satisfied with the responses and changes except for the following:

1. The title is still inappropriate. I suggest "Monolithically integrated laser and OAM emitter in InP" or something similar.
2. The way the laser wavelength is aligned to the emitter is simply by making many devices and hoping one has close alignment. This device is just very far from practical use. It has a wavelength alignment challenge, very low output power, reflections into the laser, and emits two modes simultaneously. Since the main point of the paper is integration, not solving some of the main challenges of the integration makes me feel this paper is premature. I do commend the authors for solving the substrate leakage issue.

I feel that this paper would be a great paper in an ordinary journal. Unfortunately I feel it is still not appropriate as a Nature paper. Perhaps if the authors demonstrate data transmission or something similar that is a little less premature. It's of course acceptable to publish premature results but in that case the amount of novelty must be very high and that is not the case here.

Reviewer #3 (Remarks to the Author):

I still think the title is misleading. It claims more than this paper demonstrates. It has to clearly state that it is a "vortex emitter integrated with a DFB" and NOT an OAM laser. This kind of "overselling" misleads scientists and prevents those who will truly demonstrate an electrically pumped OAM laser from publishing it!

Response to Reviewers' comments

Reviewer #1

The authors have done an excellent job of addressing all the comments and concerns of this reviewer. They provide a convincing explanation of the utility of their device in OAM communications. They also provide convincing explanations for the origin of the high losses in their current device. In summary, the device implementation is interesting, the utility of the device is potentially high. However, conceptually it is the combination of two known devices - so whether a first attempt with high losses, such as this one, but with promising and convincing routes for device performance optimisation, suffices for publication in NatComm... or if the authors should come back with a device with far improved (loss) performance... is a decision left to the editor - this reviewer is happy with either decision.

Answer: As a proof-of-concept demonstration, our device still suffers from relatively high loss. Nevertheless, our work has shown the potential of an all-in-one electrically pumped OAM emitter, which will be of interest to many researchers in this field.

Optimization of the device structure and fabrication process is currently underway, so as to reduce the loss within the ring resonator, thus enhancing the power and OAM mode purity of the emitted vortex beam.

Reviewer #2

The authors have addressed all of the Reviewers' comments. I am satisfied with the responses and changes except for the following:

(1) *The title is still inappropriate. I suggest "Monolithically integrated laser and OAM emitter in InP" or something similar.*

Answer: To better summarize our work, we have revised the title of our manuscript to “An InP-based vortex beam emitter with monolithically integrated laser.”

(2) The way the laser wavelength is aligned to the emitter is simply by making many devices and hoping one has close alignment. This device is just very far from practical use. It has a wavelength alignment challenge, very low output power, reflections into the laser, and emits two modes simultaneously. Since the main point of the paper is integration, not solving some of the main challenges of the integration makes me feel this paper is premature. I do commend the authors for solving the substrate leakage issue.

Answer: As a proof-of-concept demonstration, we are more concerned with confirming the effectiveness of our device configuration, and we prefer to achieve wavelength alignment in a most straightforward way. As a more elegant solution, wavelength alignment can be realized by adopting a quarter-wave-shifted DFB laser and incorporating on-chip microheaters to fine tune the resonance of the microring.

We admit that the current device still suffers from relatively low output power. Nevertheless, we have now identified the origin of high losses in our prototype devices. Optimization of the device structure and fabrication process is currently underway, so as to reduce the loss within the ring resonator, thus enhancing the power and OAM mode purity of the emitted vortex beam.

(3) I feel that this paper would be a great paper in an ordinary journal. Unfortunately I feel it is still not appropriate as a Nature paper. Perhaps if the authors demonstrate data transmission or something similar that is a little less premature. It's of course acceptable to publish premature results but in that case the amount of novelty must be very high and that is not the case here.

Answer: Our work has revealed the potential of an all-in-one electrically pumped OAM emitter. Our study could also lead to many new opportunities not available on pure silicon platform, e.g. co-integrating amplifiers and EA modulators with OAM emitters. As a result, we believe it will be of interest to a wide range of researchers in this field.

Reviewer #3

I still think the title is misleading. It claims more than this paper demonstrates. It has to clearly state that it is a "vortex emitter integrated with a DFB" and NOT an OAM laser. This kind of "overselling" misleads scientists and prevents those who will truly demonstrate an electrically pumped OAM laser from publishing it!

Answer: Following the Reviewer's suggestion, we have revised the title of our paper to “An InP-based vortex beam emitter with monolithically integrated laser,” so as to better convey the novelty of this work.